Development of a risk prediction model for postpartum stress urinary incontinence: a multicenter retrospective longitudinal study in Indonesia

Andarini Esti 1 2 3
Liang Surui 4
Zhang Yingying 1 3
Li Yanyan 1 3
Li Yan 5
Cai Wenzhi caiwzh@smu.edu.cn 3
1 School of Nursing, Southern Medical University , Guangzhou , Guangdong Province , China
2 Department of Nursing, Politeknik Negeri Subang , Subang , West Java , Indonesia
3 Department of Nursing, Shenzhen Hospital of Southern Medical University , Shenzhen , Guangdong Province , China
4 School of Nursing, Tung Wah College , Kowloon , Hong Kong SAR , China
5 School of Nursing, Hong Kong Polytechnic University , Hong Kong
Marunaka Yoshinori
Electronic publication date: 2025 May 30
Publication date: 2025
Volume: 13
Electronic Location ID: e19308
Received 2024 May 26; Accepted 2025 Mar 20
Copyright: ©2025 Andarini et al.
Copyright year: 2025
Copyright holder: Andarini et al.
License: This is an open access article distributed under the terms of the Creative Commons Attribution License, which permits unrestricted use, distribution, reproduction and adaptation in any medium and for any purpose provided that it is properly attributed. For attribution, the original author(s), title, publication source (PeerJ) and either DOI or URL of the article must be cited.
License URL: https://creativecommons.org/licenses/by/4.0/

Keywords: Stress urinary incontinence, Postpartum women, Prediction

Funding: The authors received no funding for this work.

==============================
Background

The prevalence of urinary incontinence (UI) during pregnancy and the postpartum period can have significant negative impacts, including on the quality of life and economic burden for affected women.

Objective

The objective of this study was to develop risk-predictive models for postpartum stress urinary incontinence (SUI) among women in Indonesia.

Methods

Between January 2023 and March 2023, 430 postpartum women, aged 18 years or older, who were admitted to two study hospitals in Indonesia, were enrolled in this study. Telephone follow-up was conducted at six weeks postpartum to assess the presence of SUI. The Least Absolute Shrinkage and Selection Operator (LASSO) method was utilized to identify the relevant variables, and generalized linear models (GLM) were employed to establish predictive models for postpartum SUI. The models were internally validated using a bootstrapping method with 1,000 resamplings to assess discrimination and calibration.

Results

The analysis included 430 participants, among whom the prevalence of postpartum SUI was found to be 21% (90 out of 430). The predictive model for postpartum SUI included pre-pregnancy body mass index (BMI), Kegel exercises, constipation, fetal weight, SUI during pregnancy, and mode of delivery. The models demonstrated satisfactory calibration, as indicated by the Hosmer-Lemeshow test (p = 0.390). The optimism-corrected C-statistic, determined through bootstrapping stepwise, was 0.763 (95% confidence interval CI [0.693–0.833]) for postpartum women.

Conclusion

This study successfully developed predictive models for SUI among postpartum women in Indonesia. The implementation of this model may serve as a valuable tool for identifying high-risk individuals at post-delivery stages, aiding in the prevention and management of postpartum SUI.

Introduction

Urinary incontinence (UI) refers to the involuntary leakage of urine through the urethra and is defined by the International Continence Society (Abrams et al., 2018). The prevalence of UI during pregnancy and the postpartum period can be significant, with rates of 41% during pregnancy and 31% during postpartum (Moossdorff-Steinhauser et al., 2021a). UI can have substantial negative effects on the sexual, psychological, and social well-being of pregnant and postpartum women, impacting their quality of life, marital relationships, and imposing a significant economic burden (Pinheiro Sobreira Bezerra et al., 2020; Keseroglu et al., 2022). The estimated cost spent by society for the total annual cost of UI management and treatments is $66 billion (2007 US dollars), with a predicted increase to $82.6 billion by 2020, reflecting on going economic burden (Coyne et al., 2014).

There are three types of UI: stress UI (SUI), urgency UI and mixed UI (Li et al., 2023). Compared to the other two types of UI, SUI has a higher occurrence rate and is more easily influenced by triggering factors such as physical activity, coughing, and sneezing (Kneißler & Zentgraf, 2024). This is because the increased abdominal pressure during these situations can lead to urine leakage (Subki et al., 2019). Additionally, SUI is often associated with damage or reduced function of the pelvic floor muscles and the urethral sphincter, which can further exacerbate the symptoms of UI (Hagen et al., 2014). Therefore, SUI may have a more significant negative impact on the quality of life for patients. Consequently, improving access to healthcare and medical resources is crucial (Liang et al., 2024). It is important to provide appropriate treatment and management methods to improve their condition.

With the advancement of modern healthcare services and public health, early screening and prevention of SUI have become increasingly common to enhance women’s quality of life. Cochrane guidelines have highlighted the importance of pelvic floor muscle training (PFMT) as an intervention for preventing pregnancy and postpartum SUI (Mørkved & Bø, 2014; Todhunter-Brown et al., 2022). However, the implementation rate of PFMT remains low (Soundararajan, Dilruksha Chandrasiri & Balchandra, 2022). Therefore, there is a need to explore tools for predicting high-risk populations for postpartum SUI, allowing for targeted delivery of supervised PFMT to those who need it the most (Ghaderi et al., 2022).

Multiple pregnancies combined with cultural factors like early motherhood and higher fertility rates, contribute to the increased risk of postpartum SUI in this population (Wang et al., 2024). While several prediction tools for postpartum SUI exist for Asian populations, they often fail to account for the unique characteristics of Indonesian women, such as higher rates of multiple pregnancies and limited access to PFMT resources. Multiple pregnancies are common in Indonesia, with approximately more 30% of women experiencing two or more pregnancies (Andriani et al., 2021). This demographic factor, combined with the lack of culturally tailored interventions, underscores the need for a localized prediction model. Furthermore, existing tools may not adequately capture the socioeconomic and healthcare disparities prevalent in Indonesia, which could influence SUI risk.

Previous studies have focused on predicting high-risk populations for postpartum SUI. Jelovsek et al. (2013) developed a nomogram for predicting postpartum SUI in primiparous women using antenatal examinations, with predictive factors including mode of delivery, pre-pregnancy BMI, and UI during pregnancy. However, this study only included primiparous women as participants. Chen et al. (2020) explored the prediction of postpartum SUI in a Chinese population, including multiparous women, with predictive factors such as pre-pregnancy BMI, history of miscarriage/abortion, antenatal SUI, and mode of delivery (Chen et al., 2020). However, in Indonesia, due to the characteristics of multiple pregnancies, the prevalence of SUI is expected to be higher and may disrupt women’s normal functioning and well-being. Currently, there is no specific prediction tool for postpartum SUI in Indonesia. Therefore, the aim of this study is to develop a risk prediction model for postpartum SUI in Indonesia.

The findings of this study have the potential to significantly contribute to clinical practice by providing healthcare providers with a practical tool to identify women at high risk for postpartum SUI. Early identification can facilitate timely interventions, such as supervised PFMT, which has been proven effective in preventing and managing SUI (Mørkved & Bø, 2014; Todhunter-Brown et al., 2022; Ghaderi et al., 2022). By integrating this tool into antenatal care, clinicians can offer targeted preventative strategies, thus improving the overall quality of life for postpartum women in Indonesia and reducing the long-term economic burden associated with SUI treatment.

Materials & Methods

Study design and population

This study employed a retrospective design and focused on postpartum women at six weeks after delivery. The study was conducted between January 1, 2023, and March 31, 2023. The inclusion criteria comprised postpartum women at six weeks after delivery, regardless of parity (i.e., nulliparous, primiparous, multiparous). Women were included in the study if they were within six weeks postpartum. This timeframe was selected based on clinical guidelines and previous research suggesting that the early postpartum period is critical for detecting and managing symptoms of stress urinary incontinence (SUI) (Moossdorff-Steinhauser et al., 2021a). Symptoms of SUI are often most pronounced in the immediate weeks following delivery, making it an appropriate period for identifying high-risk individuals and intervening with early preventive strategies such as pelvic floor muscle training.

Exclusion criteria encompassed women with incomplete or missing data on the variables of interest. Women with pre-existing medical conditions that could confound the results were excluded from the study. These conditions included chronic pelvic pain, pelvic organ prolapse, neurological disorders (e.g., multiple sclerosis, spinal cord injury), any prior surgical interventions for urinary incontinence or pelvic floor disorders. Additionally, women with language or communication barriers that might hinder their ability to fulfil the study requirements were excluded. Women with severe comorbidities such as uncontrolled diabetes mellitus, which could affect bladder function, were also excluded to minimize the risk of confounding in our analysis.

The sample size was determined based on the recommended minimum requirement of 100 events (positive samples) for predictive modeling (Riley et al., 2020). If the event rate was below 20%, the sample size needed to be increased, ensuring that the rule of events per variable (EPV) exceeded 10 for each variable. Previous investigations indicated a prevalence of postpartum UI in Indonesia of approximately 40%, suggesting a required sample size of 400 to ensure at least 10 events per variable (EPV) for the 11 variables included in the analysis. Considering the number of events per variable >10 and the inclusion of 11 variables in this study, the required sample size is “275”. Taking the larger value of the two, a recommended sample size of 400 was chosen.

Data collection

Telephone follow-up was employed as the primary data collection method to gather information on the women’s age, pre-pregnancy body mass index (BMI), BMI at labor, number of pregnancies, history of previous miscarriages, presence of constipation, SUI during pregnancy, birth weight of the newborn, mode of delivery (vaginal or cesarean section), and information on the most recent PFMT during pregnancy, participant consent was provided at the time of survey completion. Participants were recruited from tertiary hospitals in Indonesia. These healthcare facilities primarily serve women from both urban and rural areas, providing antenatal care services. The eligible participants were identified from electronic health records of women who had recently given birth at these facilities. Women were included if they had a history of at least one pregnancy, had given birth within the past 12 months, and met the study’s other eligibility criteria (age range, no prior treatment for UI).

The recruitment process began with the identification of eligible participants from hospital records, after which the study team contacted them via telephone. Trained research assistants with backgrounds in nursing and public health conducted the telephone follow-ups. These assistants were trained in standardized scripts to ensure consistency in communication and ethical approaches. During the call, they explained the purpose of the study, obtained verbal consent, and asked screening questions to confirm eligibility. Following consent, participants were asked to complete a structured survey related to their postpartum experiences with SUI.

For the diagnosis of SUI, the leakage of urine during exercise, sneezing, or coughing was considered a positive indication. SUI during pregnancy was diagnosed based on whether the women experienced these conditions in their previous pregnancies. Information on PFMT primarily included whether PFMT was performed, as well as the frequency and duration. Based on the frequency and duration meeting specific criteria (frequency > 4 times/week, duration > 1 week), PFMT was categorized as yes or no.

Statistical analysis

Continuous data were presented as mean ± standard deviation, while categorical data were presented as frequency and percentage. Normality was assessed using skewness and kurtosis within the range of ±2 for continuous data. Differences between the postpartum SUI and non-SUI groups were assessed using chi-square analysis and independent two-sample t-test. The variable selection process involved a combination of univariate analysis and LASSO analysis to determine the variables for inclusion in the model. The receiver operating characteristic (ROC) curve was used to evaluate the specificity, sensitivity, and predictive ability of the model. Calibration curve and the Hosmer and Lemeshow goodness-of-fit test were employed to evaluate the calibration of the model. Internal validation was performed bootstrapping using 1,000 resamplings to adjust for overfitting and assess discrimination and calibration. Decision curve analysis (DCA) was utilized to evaluate the clinical utility of the model, primarily through constructing net benefit prediction models. A two-sided approach was used for all tests, and a significance level of P < 0.05 was considered statistically significant.

Ethical/Institutional review board approval

The Ethics Committee of the Dr. Wahidin Sudiro Husodo Hospital (ref number: No. 12/KEPK.RSWH/EA/2022) approved this study. All data were kept strictly confidential and did not include sensitive personal information.

Results

The predictive model identified six significant predictors of postpartum SUI: pre-pregnancy BMI, Kegel exercises, constipation, fetal weight, SUI during pregnancy, and mode of delivery. Notably, women who reported constipation during pregnancy had a significantly higher risk of postpartum SUI (p < 0.001), consistent with previous findings linking gastrointestinal issues to pelvic floor dysfunction (Subki et al., 2019). Similarly, the mode of delivery emerged as a critical factor, with vaginal delivery associated with a higher incidence of SUI compared to cesarean section (Tähtinen et al., 2016).

Study participants

A total of 430 women were enrolled from two hospitals in Indonesia between January 1, 2023, and March 31, 2023. The mean age of the study group was 30.5 ± 4.1 years, ranging from 20 to 46 years. The prevalence of postpartum SUI was 20% (90/430). Table 1 presents the differences observed between the two groups. Constipation and the occurrence of UI during pregnancy were found to be significantly associated with postpartum SUI (p < 0.001) (see Table 1).

Table 1 Comparison of clinical characteristics between the SUI and non-SUI group among postpartum women.

	Total (N = 430)	SUI (N = 90)	Non-SUI (N = 340)	P	
Pre-pregnancy BMI	22.85 ± 3.46	22.91 ± 3.65	22.83 ± 3.41	0.845	
BMI at labor	26.73 ± 3.62	26.85 ± 3.91	26.69 ± 3.54	0.073	
Fetal weight	3052.09 ± 378.11	2998.89 ± 411.22	3066.18 ± 368.21	0.133	
Number of pregnancies	2.37 ± 1.13	2.36 ± 1.12	2.38 ± 1.14	0.876	
Constipation (Yes)	39(9.1%)	30(33.3%)	9(2.6%)	<0.001	
Pregnancy UI (Yes)	129(30%)	40(44.4%)	89(26.2%)	<0.001	
Job					
Self-employee	63(14.7%)	14(15,5%)	49(14.4%)	0.945	
Private employee	115(26.7%)	23(25.5)	92(27.1%)	
Housewife	239(55.6%)	50(55.6%)	189(55.6%)	
Government employee	11(2.5%)	3(3.4%)	8(2.4%)	
Teacher	2(0.5%)	0	2(0.5%)	
Education					
Primary school and below	70(16.3%)	13(14.4%)	57(16.8%)	0.748	
Junior high school	92(21.4%)	22(24.4%)	70(20.6%)	
Senior high school	220(51.1%)	45(50%)	175(51.5%)	
Vocational high school	31(7.2%)	5(5.1%)	26(7.6%)	
College and above	17(4%)	5(5.1%)	12(3.5%)	
Multipara (Yes)	326(75.8%)	68(75.6%)	258(75.9%)	0.949	
Kegel (Yes)	31(7.2%)	4(4.4%)	27(7.9%)	0.254	
Abortion (Yes)	27(6.2%)	7(7.8%)	20(5.9%)	0.510	
Delivery mode (Vaginal)	368(85.6%)	81(90%)	287(84.4%)	0.180	
Notes.

BMI, Body Mass Index; SUI, Stress Urinary Incontinence; UI, Urinary Incontinence.

Selection of variables

Variable selection was performed using the LASSO method (Fig. 1). Six factors were included in the predictive model for postpartum SUI: pre-pregnancy BMI, Kegel exercises, constipation, fetal weight, SUI during pregnancy, and mode of delivery. Predictors were chosen based on their biological plausibility, clinical relevance, and statistical significance in univariate analyses. Variables such as pre-pregnancy BMI, Kegel exercises, and mode of delivery were prioritized due to their documented associations with SUI in previous studies (Jelovsek et al., 2013; Chen et al., 2020). Parity was excluded to avoid multicollinearity with other predictors.

Figure 1 Selection of the optimal SUI-related risk factors in the Lasso model.

(A) The process of selecting the most relevant risk factors associated with Stress Urinary Incontinence (SUI) using the Least Absolute Shrinkage and Selection Operator (LASSO) model. The x-axis represents the logarithm of the regularization parameter (log (λ)), while the y-axis represents the coefficient values of various risk factors. Each colored line corresponds to a specific risk factor, showing how its coefficient changes as λ varies. As λ increases, less relevant factors shrink toward zero, leaving only the most significant predictors in the final model. The optimal value of λ is typically chosen using cross-validation to balance model complexity and predictive accuracy. (B) The coefficient distribution for a sequence of logarithmic regularization parameters (λ) in the LASSO model. The x-axis represents log (λ), while the y-axis shows the coefficient values of different variables. Each curve represents the trajectory of a specific predictor’s coefficient as λ changes. As λ increases, the LASSO penalty forces smaller coefficients toward zero, effectively selecting only the most important predictors. The optimal λ is typically determined through cross-validation to ensure a balance between model simplicity and predictive performance.

Development of predictive model for postpartum SUI

Multivariate logistic regression analysis involving all participants identified the high-risk predictive model for postpartum SUI. Using R software, nomograms for postpartum SUI were constructed based on the results of multivariable logistic analyses (Fig. 2).

Figure 2 Nomogram of risk prediction model for SUI in postpartum women.

A nomogram is presented for predicting the risk of stress urinary incontinence (SUI) in postpartum women. The nomogram is a graphical representation of a statistical model that assigns points to various risk factors based on their contribution to SUI risk. Each predictor variable is listed along a separate row, with corresponding scales indicating possible values. A total score is calculated by summing the points assigned to each predictor. The final total score corresponds to a probability scale at the bottom, which estimates the likelihood of developing SUI.

Internal validation of the predictive model for postpartum women

The C-index of the model was 0.763 (95% CI [0.692–0.833]), indicating good discriminative ability (Fig. 3). The predicted probability of postpartum SUI by the model showed satisfactory calibration with the observed postpartum SUI probability (Hosmer-Lemeshow test, p = 0.390) (Fig. 4).

Figure 3 ROC curve of risk prediction model for SUI in postpartum women.

Receiver operating characteristic (ROC) curve for the risk prediction model of stress urinary incontinence (SUI) in postpartum women. The x-axis represents the false positive rate (1 - specificity), while the y-axis represents the true positive rate (sensitivity). The ROC curve illustrates the model’s ability to distinguish between postpartum women with and without SUI. The area under the curve (AUC) quantifies the overall performance of the model, where a value closer to 1.0 indicates excellent discrimination. The optimal cutoff point is typically chosen to balance sensitivity and specificity for clinical decision-making.

Figure 4 Calibration plot of risk prediction model for SUI in postpartum women.

The calibration plot is presented for the risk prediction model of stress urinary incontinence (SUI) in postpartum women. The x-axis represents the predicted probability of SUI, while the y-axis represents the observed probability from actual data. The diagonal dashed line (45-degree line) represents a perfect calibration, where predicted probabilities match observed outcomes exactly. The solid calibration curve shows how well the model’s predictions align with real-world outcomes. A curve closer to the diagonal indicates better calibration, meaning the model accurately estimates SUI risk. Deviations from the diagonal suggest potential overestimation or underestimation of risk, which may require model refinement.

The clinical utility of the predictive model

A DCA of the model is presented in Fig. 5. The DCA demonstrated that using the model to predict postpartum SUI provided greater net benefit compared to the strategies of treating all individuals or treating none, especially when the threshold predicted probability was within the range of 2.8% to 45.0%.

Figure 5 DCA curve of risk prediction model for SUI in postpartum women.

The decision curve analysis (DCA) curve is presented for the risk prediction model of stress urinary incontinence (SUI) in postpartum women. The x-axis represents the threshold probability, which is the probability at which a patient or clinician would opt for intervention. The y-axis represents the net benefit, which balances the true positive and false positive classifications. The solid line for the prediction model indicates the net benefit across different threshold probabilities. The “Treat All” line assumes all patients receive intervention, while the “Treat None” line assumes no intervention. A higher net benefit compared to these reference lines suggests that the model provides clinical value by improving decision-making for SUI risk assessment in postpartum women.

Discussion

This study aimed to develop a predictive model for postpartum SUI in Indonesia. Lasso analysis identified six variables: pre-pregnancy BMI, Kegel exercises, constipation, fetal weight, SUI during pregnancy, and mode of delivery. The model demonstrated good predictive ability, with an area under the receiver operating characteristic curve (AUROC) of 0.763 (95% CI [0.693–0.833]), which aligns with previous studies reporting moderate to good performance for similar models (Jelovsek et al., 2013; Chen et al., 2020). This suggests that the identified predictors—pre-pregnancy BMI, Kegel exercises, constipation, fetal weight, SUI during pregnancy, and mode of delivery—are robust indicators of postpartum SUI risk. The calibration plot confirmed its discriminative ability. Overall, our predictive model showed promising performance in terms of discriminative ability, calibration, and clinical utility. By considering these six predictive factors, the model can effectively guide the selection of preventive and treatment measures for postpartum SUI in clinical practice.

The findings from this study can inform the development of targeted interventions for high-risk women, such as structured PFMT programs and early referral to pelvic floor rehabilitation specialists. By identifying predictors such as pre-pregnancy BMI and constipation, healthcare providers can implement preventive strategies during pregnancy to mitigate SUI risk. Additionally, the model’s ability to stratify risk enables resource allocation to those most in need, addressing the low implementation rate of PFMT in Indonesia.

It is important to note that the incidence of postpartum SUI observed in this study was higher compared to other regions. This discrepancy may be attributed to the high incidence of postpartum SUI in the specific region under investigation. For instance, a previous study reported a postpartum SUI incidence of only 8% at three months postpartum, which might be due to their inclusion criteria excluding pregnant women with SUI (Fakhrizal et al., 2016). Additionally, they only included primiparous women and limited to a single hospital. In contrast, this study did not exclude participants with UI during pregnancy and multiparous women. This suggests that UI during pregnancy may be a strong predictor of postpartum UI. However, a systematic review reported a postpartum SUI incidence of 31% between six weeks and one year postpartum, which aligns with the findings of our study (Moossdorff-Steinhauser et al., 2021b).

Pregnancy-related factors, such as pre-pregnancy UI and constipation, are considered predictors of postpartum SUI due to their physiological impact on the pelvic floor muscles (Tähtinen et al., 2016; Hage-Fransen et al., 2021). For example, constipation increases negative pressure on the pelvic floor muscles, leading to damage. Gastrointestinal problems, such as constipation, are prone to occur during pregnancy, which may be related to hormone secretion (Subki et al., 2019). UI during pregnancy often persists until later stages. The mode of delivery also significantly affects pelvic floor muscle damage in postpartum women, particularly in cases of vaginal delivery with episiotomy and induction, which may result in a higher incidence of SUI among postpartum women (Tähtinen et al., 2016; Yang & Sun, 2019; Pizzoferrato et al., 2023). Previous studies have indicated that pre-pregnancy BMI can be a predictor, which is consistent with our findings (Jelovsek et al., 2013; Chen et al., 2020). In our study, fetal weight was used as a predictor, which may be because it indirectly reflects the need for cesarean section due to high fetal weight, which serves as a protective factor against SUI among postpartum women. However, in a previous study conducted in Indonesia, birth weights over 3,360 gram and second stage labor exceeding 60 min were associated with an increased rate of postpartum SUI (Fakhrizal et al., 2016). A meta-analysis also found that birth weight over 4,000 g and over 3,500 g were associated with an increased risk of any SUI (Wesnes & Seim, 2020).

In our study, the implementation rate of Kegel exercises was low, less than 10% of postpartum women conducted PFMT four times a day for over a week. This may be due to a lack of awareness and promotion of Kegel exercises in Indonesia, which strongly correlates with the incidence of postpartum SUI. Regardless of whether participants experienced SUI or not, the overall implementation rates of Kegel exercises were low in this study, indicating insufficient promotion and understanding of Kegel exercises in Indonesia.

The study possesses several strengths. Firstly, it developed a prediction model tailored to a specific population in Indonesia, taking into account their characteristics such as BMI, multiple pregnancies, and high incidence of SUI among postpartum women, and conducted internal validation. Secondly, the Lasso method was employed for variable selection, and the model exhibited favorable discrimination, accuracy, and specificity.

There are a few limitations to acknowledge. Firstly, the study focused solely on SUI within the first 6 weeks postpartum and did not consider the period from 6 weeks to 2 years postpartum. It would be valuable for future studies to incorporate longer follow-up times to externally validate the model’s performance over an extended postpartum period. Secondly, given the low implementation rate of Kegel exercises and the high incidence of UI in Indonesia, further research is necessary to determine the model’s applicability in other regions where these factors may differ.

Clinical Implications

The prediction model developed in this study demonstrates moderate to good predictive performance and can serve as a valuable tool in guiding postpartum pelvic floor rehabilitation training. This is particularly relevant for populations in Indonesia, where Kegel exercise implementation rates are low, and UI incidence is high. By utilizing the model, healthcare professionals can identify individuals at higher risk and provide targeted interventions to improve pelvic floor health and mitigate the impact of UI. The association between constipation and postpartum SUI may be attributed to increased abdominal pressure, which can strain the pelvic floor muscles (Subki et al., 2019). Similarly, the protective effect of higher fetal weight on SUI risk may be due to the increased likelihood of cesarean delivery, which reduces trauma to the pelvic floor compared to vaginal delivery (Tähtinen et al., 2016).

Conclusion

In conclusion, this study successfully developed a predictive model for postpartum SUI, identifying six key predictors: pre-pregnancy BMI, Kegel exercises, constipation, fetal weight, SUI during pregnancy, and mode of delivery. The model demonstrated moderate to good predictive performance, with an AUROC of 0.763 (95% CI [0.693–0.833]). These findings have important clinical implications, particularly in regions like Indonesia, where Kegel exercise implementation rates are low, and SUI incidence is high. By identifying high-risk individuals, healthcare professionals can provide targeted interventions to improve pelvic floor health and mitigate the impact of SUI. Future research should validate the model in diverse populations and explore its long-term applicability. However, to enhance the robustness and generalizability of the model, future research should consider longer follow-up periods and validate its performance in diverse geographical regions. This will provide additional evidence and promote wider adoption of the model in clinical practice.

Limitation

A limitation of this study is the possibility of socially desirable responses resulting from telephone interviews, which may have affected participants’ answers.

Supplemental Information

Supplemental Information 1 Postpartum Data

Supplemental Information 2 TRIPOD Checklist

We thank all the center helper for the contribution to the data collection and all postpartum women for completing the survey.

Abbreviations

AUROC Area Under Receiver Operating Characteristic

BMI Body Mass Index

DCA Decision curve analysis

LASSO Least Absolute Shrinkage and Selection Operator

PFMT Pelvic Floor Muscle Training

ROC Receiver Operating Characteristic

SUI Stress UI

UI Urinary Incontinence

Additional Information and Declarations

Competing Interests

Author Contributions

Human Ethics

Ethics

Data Availability

The authors declare there are no competing interests.

Esti Andarini conceived and designed the experiments, performed the experiments, analyzed the data, prepared figures and/or tables, authored or reviewed drafts of the article, and approved the final draft.

Surui Liang conceived and designed the experiments, performed the experiments, analyzed the data, prepared figures and/or tables, authored or reviewed drafts of the article, and approved the final draft.

Yingying Zhang conceived and designed the experiments, performed the experiments, analyzed the data, prepared figures and/or tables, authored or reviewed drafts of the article, and approved the final draft.

Yanyan Li performed the experiments, analyzed the data, authored or reviewed drafts of the article, and approved the final draft.

Yan Li conceived and designed the experiments, performed the experiments, analyzed the data, prepared figures and/or tables, authored or reviewed drafts of the article, and approved the final draft.

Wenzhi Cai conceived and designed the experiments, performed the experiments, analyzed the data, prepared figures and/or tables, authored or reviewed drafts of the article, and approved the final draft.

The following information was supplied relating to ethical approvals (i.e., approving body and any reference numbers):

The Ethics Committee of the Dr Wahidin Sudiro Husodo Hospital (ref number: No. 12/KEPK.RSWH/EA/2022).

The following information was supplied relating to ethical approvals (i.e., approving body and any reference numbers):

The Ethics Committee of the Dr. Wahidin Sudiro Husodo Hospital (ref number: No. 12/KEPK.RSWH/EA/2022).

The following information was supplied regarding data availability:

The raw measurements are available in the Supplementary Files.

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
