# Peer review of "Development of a risk prediction model for postpartum stress urinary incontinence: a multicenter retrospective longitudinal study in Indonesia"

_PeerJ, doi:10.7717/peerj.19308_

## Round 0.1 · original submission · Major Revisions

If you feel you can revise your manuscript according to the reviewers' comments, please revise your manuscript and submit it. Please also send us your written responses to each of the reviewers' comments.

Yours,

Yoshi

Prof. Yoshinori Marunaka, M.D., Ph.D.

Reviewer 1 ·

Basic reporting

Introduction

Line 54-56 It might be better to update the cited reference, as the current one is quite dated.
Line 57-61 References are needed

The introduction section provides detailed background information on SUI. However, the significance and necessity of conducting the current study should be further expanded. You mentioned that multiple pregnancies are common in Indonesia; is this the sole reason for developing a new prediction model in this country? There are already tools developed for Asian women. Perhaps more clarification is needed. Additionally, some prevalence data on multiple pregnancies could be included.

How could the findings from the current study contribute to clinical practice? This could be briefly introduced in the introduction section.

Materials and methods
The data collection procedures should be more detailed. For example, who conducted these telephone follow-ups? How did you screen, identify, and approach these eligible women? if women were recruited from a specific setting, then a brief introduction of the setting should be provided.


Discussion
There could be more discussion on how the clinical practice could be improved based on the findings from this study.

Experimental design

How did you select the predictors for SUI/baseline maternal characteristics collected? Parity seems to be a possible factor associated with SUI; why this was not considered a possible predictor?

Please specify the rationale for including women within six weeks postpartum.

Exclusion criteria: “pre-existing medical conditions that could confound the results”, this should be more specific. What are these conditions?

Validity of the findings

No comment

Additional comments

No comment

Reviewer 2 ·

Basic reporting

This manuscript reported that has as aim “to develop a risk prediction model for postpartum SUI in Indonesia”.
- There were editorial errors and there were some odd sentence structures in the manuscript. Professional English editing and proofreading should help improve the clarity of this manuscript; for example the such as the use of acronyms sometimes and not others(i.e line 58 urinary incontinence, before UI, line 57; pelvic floor muscle training (PFMT) line 68, and pelvic floor muscle training, line 70.
In line 33 to correct the word iterations. Please, let the authors carefully review the wording and use of acronyms throughout the text. Professional English editing and proofreading should help improve the clarity of this manuscript.

- It is important that the discussion be reviewed and the results found are substantiated.

- Regarding the figures, it is necessary to review the quality and improve the description of the figure captions.

Experimental design

- Please describe in more detail the questions related to Telephone follow-up.

- Could you explain how you determined the sample size?

Validity of the findings

The results are important, however, it is necessary to describe them in greater detail; because they are very summarized.

It is necessary that the figures be of better quality.


First, make the conclusion according to your results, followed by your possible applications of the model.

Additional comments

The model they propose is interesting for the population. However, the authors must do a thorough review of the writing.

In the discussion they can detail the physiological implications of pelvic floor exercises.

Annotated reviews are not available for download in order to protect the identity of reviewers who chose to remain anonymous.

---

## Round 0.2 · Minor Revisions

Please revise your manuscript according to the reviewer's comments, and resubmit it.
Yours,
Yoshi
Prof. Yoshinori Marunaka, M.D., Ph.D.

Reviewer 1 ·

Basic reporting

Thank you for your time and careful consideration in addressing my feedback. The manuscript has been significantly improved, and I only have a few minor suggestions.

Line 62 There is an error in the in-text citation
Lines 133: Please add references to support this statement ‘’ Early identification can facilitate timely interventions, such as supervised pelvic floor muscle training (PFMT), which has been proven effective in preventing and managing SUI.
Lines 177 ‘tertiary hospitals’
Line 160: Please double check the revised manuscript. Do the communication and language barriers fall under pre-existing medical conditions?

Experimental design

Lines 189: Please clarify how participants completed the structured survey? Did the research assistants ask them the survey questions one by one? If so, this may lead to potential bias, such as socially desirable responses, and should be acknowledged as a limitation.

Validity of the findings

No comment

Additional comments

No Comment

---

## Round 0.3 · accepted · Accept

Congratulations.
Yours,
Prof. Yoshinori Marunaka. M.D., Ph.D.